# The Importance of Carbapenemase-Producing Enterobacterales in African Countries: Evolution and Current Burden

**DOI:** 10.3390/antibiotics13040295

**Published:** 2024-03-24

**Authors:** Edgar-Costin Chelaru, Andrei-Alexandru Muntean, Mihai-Octav Hogea, Mădălina-Maria Muntean, Mircea-Ioan Popa, Gabriela-Loredana Popa

**Affiliations:** 1Department of Microbiology II, Faculty of Medicine, Carol Davila University of Medicine and Pharmacy, 020021 Bucharest, Romania; edgar-costin.chelaru@drd.umfcd.ro (E.-C.C.); alexandru.muntean@umfcd.ro (A.-A.M.); mihai-octav.hogea@drd.umfcd.ro (M.-O.H.); madalina.muntean@umfcd.ro (M.-M.M.); 2Department of Microbiology, Cantacuzino National Military Medical Institute for Research and Development, 050096 Bucharest, Romania; 3Department of Microbiology, Faculty of Dentistry, Carol Davila University of Medicine and Pharmacy, 020021 Bucharest, Romania; gabriela.popa@umfcd.ro; 4Parasitic Disease Department, Colentina Clinical Hospital, 020125 Bucharest, Romania

**Keywords:** Africa, Enterobacterales, carbapenem resistance, carbapenemase, CPE, colonization

## Abstract

Antimicrobial resistance (AMR) is a worldwide healthcare problem. Multidrug-resistant organisms (MDROs) can spread quickly owing to their resistance mechanisms. Although colonized individuals are crucial for MDRO dissemination, colonizing microbes can lead to symptomatic infections in carriers. Carbapenemase-producing Enterobacterales (CPE) are among the most important MDROs involved in colonizations and infections with severe outcomes. This review aimed to track down the first reports of CPE in Africa, describe their dissemination throughout African countries and summarize the current status of CRE and CPE data, highlighting current knowledge and limitations of reported data. Two database queries were undertaken using Medical Subject Headings (MeSH), employing relevant keywords to identify articles that had as their topics beta-lactamases, carbapenemases and carbapenem resistance pertaining to Africa or African regions and countries. The first information on CPE could be traced back to the mid-2000s, but data for many African countries were established after 2015–2018. Information is presented chronologically for each country. Although no clear conclusions could be drawn for some countries, it was observed that CPE infections and colonizations are present in most African countries and that carbapenem-resistance levels are rising. The most common CPE involved are *Klebsiella pneumoniae* and *Escherichia coli*, and the most prevalent carbapenemases are NDM-type and OXA-48-type enzymes. Prophylactic measures, such as screening, are required to combat this phenomenon.

## 1. Introduction

The issue of antimicrobial resistance (AMR) in healthcare is intricate, dynamic and ever-evolving globally [1,2]. Although resistance to antiviral, antifungal and antiparasitic medications poses significant challenges, bacterial resistance to antibiotics and chemotherapeutics seems to be the most troublesome, as bacterial infections are ubiquitous and extremely diverse. Resistance develops and spreads rapidly in different fields of activity, including human and veterinary medicine and the food industry [1,3,4,5,6].

Although antimicrobial-resistant organisms are known to cause severe healthcare-associated infections, such bacteria are increasingly more common in community-acquired infections [7,8]. The silent spread of multidrug-resistant organisms (MDROs), such as carbapenem-resistant organisms (CROs), including carbapenemase-producing Enterobacterales (CPE), extended spectrum β-lactamase (ESBL)-producing Enterobacterales, methicillin-resistant *Staphylococcus aureus* (MRSA), vancomycin resistant *Enterococcus* (VRE) and others, in carriers is concerning, as they disseminate in and between healthcare institutions and communities and even across borders and continents (e.g., via travelers, diasporas and migrants) [7,8,9,10,11,12,13,14,15,16,17]. It has been demonstrated that chronic carriers are prone to developing severe, hard-to-treat infections themselves, with significant morbidity and mortality rates, as some colonizing bacteria tend to express their pathogenicity factors and become virulent in particular circumstances, such as immunosuppression, imbalance of the bacterial flora, trauma, surgery, antimicrobial treatment, etc. [9,18,19,20,21,22,23,24].

Many bacterial genera and species presenting with varied mechanisms of resistance have been described, and new mechanisms are constantly being discovered [25]. Mutations in genes encoding for structural proteins can lead to different adaptive modifications, such as permeability decrease, while acquisition and regulation of genes can lead to the development of efflux pumps and a decrease in the number of porins, respectively. However, enzymatic mechanisms are the most concerning, as their encoding genes are frequently located on mobile, transposable elements, which can be easily transmitted between bacteria, not only to descendants but also horizontally, between different strains, sometimes even species or genera [1,5,26,27].

Some of the most important enzymes associated with Enterobacterales are β-lactamases, which can inactivate various numbers of β-lactam antibiotics. β-lactam antibiotics are very important therapeutic resources because of their bactericidal effects, often representing the first and the most effective treatment choice. Among β-lactamases, extended spectrum β-lactamases (ESBLs), cephalosporinases (especially AmpCs) and carbapenemases are the most significant [12,26,27]. Of these, carbapenemases are enzymes that can render most of, or the entire, β-lactam group unsuitable for treatment and, in association with other mechanisms, can lead to the emergence of multidrug-resistant (MDR), extended drug-resistant (XDR) and pandrug-resistant (PDR) strains [27,28].

Although some reviews addressing the topic of antimicrobial resistance or CPE prevalence based on data obtained in African countries or regions were identified, to our knowledge, this is the first study containing the most recent available data on the African continent, including CPE carriage data. This review sought to identify the first studies reporting the emergence of CPE in African countries and determine how these microorganisms spread and to offer a general picture regarding the current situation of CRE and CPE. Due to the fact that reports on some African regions or countries were scarce, we have highlighted current knowledge and the limitations of the reported data.

## 2. Results

The number of studies identified from the first database search was 71. After eliminating duplicates (*n* = 4), the papers were evaluated based on their titles and abstracts. In order to narrow our results, we excluded papers that provided descriptions of other Gram-negative bacilli (*n* = 11), as well as those studies that focused on resistance mechanisms that affected susceptibility to carbapenems other than carbapenemases (*n* = 2) and those not written in English (*n* = 1).

Subsequently, the remaining papers (*n* = 53) were examined in their entirety. Studies that did not include data in regard to CPE and/or CRE (*n* = 5), studies with unclear methodologies in regard to the processing of clinical samples (*n* = 7) and studies that did not discuss African countries (*n* = 4) were further excluded.

The number of studies included after the first database search was 37.

A similar approach was taken for the second database search. The number of works identified through the use of our keywords was 309. After the removal of duplicates (*n* = 15), 294 papers were evaluated based on their titles and abstracts. Following the aforementioned exclusion criteria, we excluded papers that provided descriptions of other Gram-negative bacilli (*n* = 32), as well as those studies that focused on resistance mechanisms that affected susceptibility to carbapenems other than carbapenemases (*n* = 30) and those not written in English (*n* = 5).

Two hundred and twenty-seven manuscripts were then evaluated in their entirety. Studies that did not include data in regard to CPE and/or CRE (*n* = 52), studies with unclear methodologies in regard to the processing of clinical samples (*n* = 13) and studies that did not discuss African countries (*n* = 58) were further excluded.

The number of studies included after the first database search was 104.

Our investigation comprised a total of 141 studies. A flowchart of the selection process can be found in Figure 1.

Although CRE and even CPE might have been reported in Africa before, the authors of a 2010 article tracked down and published the first documented case of *Klebsiella pneumoniae* NDM-1 infection in Africa, which originated in Kenya in 2007 [29]. The strain had a similar pulsed-field gel electrophoresis pattern to the one first reported in 2008 in Sweden in a patient previously hospitalized in India [29]. Another study, also published in 2010, described two strains of *Escherichia coli* and *Klebsiella pneumoniae* isolated from Algerian patients in 2008, in which a novel VIM carbapenemase, VIM-19, was recorded [30].

Some of the first cases of CR and CP non-Enterobacterales were reported in South Africa even sooner: a study published in 2001 described a case of *Pseudomonas aeruginosa* harboring GES-2 with increased hydrolyzing activity with respect to imipenem that was isolated from blood cultures [31], while another study published in 2005 described infections caused by *Acinetobacter baumannii* OXA-23—the authors suspected the emergence of these strains to have occurred in 2002 [32]. Later, in 2008 and 2010, such strains isolated in 2005–2006 were also reported in Tunisia and Madagascar [33,34].

These findings suggest that CP microorganisms reached African countries a few years after they were first identified and described in GNB [25,35,36,37]. In the following years, CPE have been reported and described with an increasing frequency in many African healthcare units, in infected patients and carriers.

**Algeria:** The first report regarding CPE-infected Algerian patients was made in 2010 (described above) [30]. In a 2014 study, *E. coli* producing OXA-48 enzymes, sampled from a urinary tract infection (UTI) in 2012, were reported for the first time in Algeria [38]. Later, in 2015, OXA-48 or NDM-5 *E. coli* were reported in 5 of 200 (2.5%) pets screened for intestinal carriage [39]. A 2016 publication reported 14 carbapenemase-producing organisms (CPOs) (OXA-48, NDM and OXA-23) among 32 carbapenem-resistant organisms (CROs) isolated from clinical samples and surfaces. Two of them were the first *Enterobacter cloacae* strains with OXA-48 encoding genes (*bla*_OXA-48_) reported in Algeria, while the other three were *bla*_OXA-48_
*K. pneumoniae* [40], while, in another study, among 186 GNB from clinical isolates, 161 were *Enterobacteriaceae*, 36 were CR-GNB and 2 were *bla*_OXA-48_
*K. pneumoniae* (1.2% CPE prevalence among *Enterobacteriaceae*) [41]. A 2017 study reported that, among 99 GNB isolated in 2014–2015 from stool samples and surfaces, 10 were CR-CPOs. Two were *bla*_OXA-48_
*Enterobacteriaceae* (one *E. coli* and one *K. pneumoniae*). The other eight were *Acinetobacter* spp. (seven *A. baumannii* and one *A. nosocomialis*), among which four of the *A. baumannii* and the *A. nosocomialis* were *bla*_NDM-1_ and the remaining three *A. baumannii* were *bla*_OXA-23_ [42]. A 2020 publication reported that among 42 colorectal cancer patients from 2019 screened for CPE fecal carriage, 1 patient was carrying OXA-48-producing *K. pneumoniae* [43]. In 2022, a strain of NDM-5-producing *K. pneumoniae* isolated in 2017–2018 was reported [44]. Overall, data on human CPE carriage are still scarce for Algeria, but CPE prevalence varied in studies from 1.2% to 2.5%.

**Angola:** A 2016 publication reported that, following a 2015 screening for CPE rectal colonization (rectal swabs were collected), 48/157 children (27.4%) carried Enterobacterlaes encoding for OXA-181 (an OXA-48-like enzyme) or NDM-1 [45]. This study was followed by another one, published by the same authors in 2018, where increased rates of CPE were reported (28/36 screened patients) and the emergence of NDM-5 was noted [46]. 

**Benin:** In a 2023 study, *bla*_GES_ genes were identified in hospital wastewater and in water intended for handwashing [47]. Another 2023 study which evaluated 390 urine samples from 2021–2022 isolated 103 *Enterobacteriaceae* (*E. coli*, *Serratia* spp., *Klebsiella* spp., *Citrobacter freundii* and *Enterobacter intermedius*). Although a low imipenem resistance rate was observed in 27.18% strains, no data on CP rates are available [48].

**Burkina Faso:** In a 2023 study, *bla*_GES_, *bla*_IMP_, *bla*_NDM_, *bla*_OXA-48-like_, *bla*_OXA58-like_ and *bla*_VIM_ genes were identified in Burkina Faso hospital wastewater [47]. Another 2023 study which evaluated 170 *E. coli* and *K. pneumoniae* strains isolated from 82/84 healthcare center wastewater samples identified 10 CPE, of which 6 were NDM, 3 were OXA-48 producers, and 1 was an NDM + OXA-48 co-producer [49].

**Botswana:** Relevant data were found in a 2021 study that evaluated CRE intestinal colonization prevalence (rectal swabs were collected). Of 2469 participants recruited from different environments (hospitals, clinics and communities), 42 were colonized with CRE and 10 were colonized with multiple strains. The CRE species were *E. coli, K. pneumoniae* and *E. cloacae*. Of all the hospital subjects, 6.8% were colonized, while in clinics and communities only 0.7% and 0.2% tested positive for CRE [50].

**Cape Verde:** A study published in 2022 showed that 6 of 98 patients screened with rectal swabs carried *E. coli* and *K. pneumoniae* encoding for OXA-48-like enzymes [51].

**Djibouti:** A 2023 study revealed a prevalence of 1.9% CP-GNB (32/1650) among all samples and 1.2% CP-GNB (25/1300) among human samples. The samples were collected from multiple sites: 1300 were collected from humans (800 from communities and 500 from hospitals), and the others were collected from animals, fish and water. Among the 32 bacterial isolates identified, 19 were *E. coli*, 5 were *K. pneumoniae* and 1 was *Proteus mirabilis* (25 CPE, 1.5% prevalence) associated with *bla*_NDM_, *bla*_OXA-48_ and *bla*_OXA-181_ [52].

**Egypt:** A study from 2012 described one of the first infections caused by *K. pneumoniae* producing NDM-1 [53]. A 2018 study reported MBL-producing *Serratia marcescens* (VIM-2 and IMP-4) isolated from intensive care unit (ICU) patients in Cairo [54]. A 2019 study reported that out of 413 *Enterobacteriaceae* isolated from cultured rectal swabs (2015–2016), 100 (24%) were CRE. Eighty percent (80%) of CRE were CPE (19.4% overall CPE). *bla*_OXA-48_ and *bla*_NDM-1_ were the most prevalent genes, while *E. coli* and *Klebsiella* spp. were the most prevalent species [55]. A 2020 study reported an *E. coli* NDM isolated from a patient with diarrheal disease [56]. A study published in 2023 reported, among 150 isolates from 2019, 30 CR-GNB (20%), of which 26 (17.33%) were CRE. *K. pneumoniae* was the most prevalent CR species (10/30), and *bla*_NDM_ was the most prevalent gene (15/30), frequently found in plasmids. Twenty-one out of the thirty CR-GNB (21/30) harbored CP genes. Of these, 19 (12.66%) were Enterobacterales and 2 were *P. aeruginosa*. Other CPE were *E. coli, E. cloacae* and *Citrobacter freundii*, while other genes were represented by *bla*_VIM_, *bla*_IMP_ and *bla*_KPC_ [57]. Another 2023 study described 150 Enterobacterales strains isolated from clinical samples (2019–2020), out of which fifty-three (53/150) were deemed CR by antimicrobial susceptibility (AST) screening and confirmed as CPE by molecular methods (35.33% CPE prevalence). Genotypically, 30/53 isolates carried *bla*_NDM-1_ and 41/53 carried *bla*_OXA-48_ (18 isolates carried both genes). *K. pneumoniae* was the most prevalent (37/53), followed by *E. coli* (15/53) and *K. oxytoca* (1/53) [58]. Overall, the reported CPE prevalence ranged from 12.66% to 35.33%, but data on CPE colonization are still scarce for Egypt (one report of 19.4% was found).

**Ethiopia:** In 2016, KPC and MBL *K. pneumoniae* strains isolated in 2012 from two colonized children (stool samples/rectal swabs) were reported among 267 sampled patients (154 adults and 94 children), resulting in 0.75% CPE intestinal carriage [59]. Larger studies reported prevalences of 2.73% (2015) [60] and 2% (2019) [61] and even 12.2% CPE (2017) among isolated Enterobacterales (although it is not clear if the isolates were CPE or if CRE with other resistance mechanisms were also included) [62]. One study identified 16.2% CPOs among 185 GNB isolated from 532 samples (2019); it must be noted that the prevalence was calculated and reported to be 148 MDR-GNB; the true prevalence would be 13% CPOs among GNB, 12.4% CPE among 185 GNB and 4.3% CPE in the studied population [63]. In a 2021 study, 17 of 312 Enterobacterales isolated from clinical samples were potential CPE and 8 (2.6%) were phenotypically confirmed by mCIM. The eight strains were *K. pneumoniae* (four), *E. coli* (three) and *Enterobacter* spp. (one); further testing revealed the presence of OXA-48, MBL and KPC + OXA-48 [64]. In another 2021 study which screened 833 subjects (various clinical samples), 141 GNB were isolated and 51 proved to be MDR. Eight passed as CPE (*Enterobacter* spp., *Klebsiella* spp. and *E. coli*) according to the Modified Hodge Test (MHT), resulting in an approximately 1% CPE prevalence [65]. Many studies were published in 2022. In one of them, 301 *Enterobacteriaceae* isolated from 1416 patients were analyzed, ~7% (20/301 strains, *K. pneumonia* and *E. cloacae*) of which carried *bla*_NDM_ and/or *bla*_OXA-48_ genes [66], while, in another study, 8% of isolated *Enterobacteriaceae* were CR, with 6% confirmed by mCIM as CPE (*E. cloacae, K. pneumoniae* and *E. coli*) [67]. One study showed that, out of 290 stool samples collected from asymptomatic food handlers, 7 (2.4%) tested positive for CPE presence, especially *E. coli* and *K. pneumoniae* [68]. Another article reported that, out of 132 *K. pneumoniae* strains isolated from patients in previous years, 39 (29.6%) were CR and 28 (21.2%) were CPE. Twenty-six harbored *bla*_NDM_, of which one co-harbored *bla*_KPC_ [69]. A 2023 study revealed that of 183 diarrheal pathotype *E. coli* isolated from children, 4 (2.2%) were CPE [70], while another study evaluating GNB isolated from blood cultures revealed prevalences of 25.1% for CP-GNB and 5.6% for MBL producers among 231 GNB (179 Enterobacterales), with 2% CPE in the studied population [71]. A systematic review from 2023 reported an overall 5.44% pooled prevalence of CPE in Ethiopia, ranging from 2.24% in 2015–2016 to 17.44% in 2017–2018 and from 1.65% in the southern region to 6.45% in Central Ethiopia [72].

**Gabon:** A 2022 screening study evaluated 98 Enterobacterales isolated from diarrheal stools and reported 28 CRE [73]. In 2023, data on CP-GNB collected from 2016 to 2018 were published. A total of 14/869 clinical isolates (1.61%) and 1/19 fecal samples (carriage) presented CP-GNB, with higher rates among inpatients (2.98%) than outpatients (0.33%). The most prevalent GNB were *K. pneumoniae* (8/15) and *A. baumannii* (4/15), and the most prevalent gene was *bla*_OXA-48_, followed by *bla*_NDM-5_. Regarding Enterobacterales, 10/869 clinical isolates (1.15%) were CPE, in addition to 1 isolate from stool samples tested for carriage [74].

**Ghana:** In a study published in 2019, 26 out of 111 CR-GNB (including 7 CRE) isolated in 2012–2014 presented NDM-1, OXA-48 and VIM-1 genes (VIM-1 was found only in *Pseudomonas* spp.) [75]. In a 2020 published study, MDR-GNB carriage (ear, axilla, groin and perianal swabs) was evaluated in 228 hospitalized neonates recruited from neonatal ICUs (NICUs). Two hundred and seventy-six (276) GNB were isolated from 175 positive patients, of which 115 were *Klebsiella* spp. A total of 18/115 (15.6%) *Klebsiella* spp. expressed CR and harbored *bla*_OXA-181_. Sixteen of two hundred and twenty-eight (16/228, 7%) neonates developed GNB bloodstream infection, and in two of them sequencing confirmed that the colonizing MDROs were responsible. The confirmed CPE carriage was ~10% [76]. In a study from 2022, 26 strains harboring *bla*_NDM_ and 1 strain harboring *bla*_OXA-48_ were isolated from 231 hospital surfaces. One strain was *K. pneumoniae*, and the rest were *Acinetobacter* spp. [77]. Another study from 2022 revealed that 57 GNB were isolated from 410 nasopharyngeal samples (swabs) collected in 2016 from small children, especially *E. coli*, *K. pneumoniae* and *E. cloacae*. Among the 57 strains, 5 tested positive as carbapenemase producers according to the MHT (3 *E. coli,* 1 *K. pneumoniae* and 1 *Acinetobacter* spp.). Nasopharyngeal CPO carriage was found in 1.46% of screened children, while nasopharyngeal CPE carriage was found in 0.97% [78]. In a 2023 study, 181 GNB isolated from clinical samples were processed and 161 were identified as Enterobacterales. Among the 161 Enterobacterales, 31 were CRE but only 4 encoded carbapenemases: 1 *bla*_OXA-48_ + *bla*_KPC_
*E. coli,* 1 *bla*_OXA-48_ + *bla*_KPC_
*K. pneumoniae*, 1 *bla*_NDM_ *K. pneumoniae* and 1 *bla*_NDM_ *Providencia vermicola*. This equaled a CPE prevalence of 2.2–2.5% [79]. A 2023 study that evaluated stool samples for MDR colonization showed that, out of 736 healthy residents, 2 (0.3%) participants carried *bla*_NDM-1_
*E. coli* [80]. Thus, the reported CPE carriage prevalence across studies ranged from 0.3% to 10% in Ghana.

**Kenya:** The first report regarding CPE in Kenya was dated 2010 (described above) [29]. In an article from 2020, OXA-48 *Salmonella* isolated from a Kenyan patient with diarrheal disease was reported [56]. A study published in 2022 analyzed 89 *K. pneumoniae* strains isolated between 2015 and 2020 in Kenya and described 2 strains (2.24%) harboring *bla*_NDM-1_ and *bla*_OXA-181_ [81]. Another 2022 publication reported screening data from 2019: 300 mothers and their newborn babies were evaluated for MDR-GNB colonization. Two percent (2%) of mothers (*n* = 7/300) had CROs (CRE) isolated from vaginal secretions. For newborns, a 3% (*n* = 8/300) CROs (CRE) rate was observed on admission and a fivefold increase was recorded (up to 14%, *n* = 29/218) upon discharge. Among the CROs, the most prevalent were *K. pneumoniae* and *E. coli* harboring *bla*_NDM-1_, *bla*_NDM-5_ and *bla*_NDM-7_, but *bla*_OXA-181_ and *bla*_OXA-232_ were also identified. Furthermore, a 3% (*n* = 3/164) CRO (CRE) rate was reported in the hospital environment [82]. A surveillance report published in 2023 evaluated 119 stool samples and rectal swabs collected from 42 infants in 2018–2019. In total, 18 infants were from Kenya and 24 were from Nigeria. Seven of eighteen (7/18) Kenyan infants tested positive for CPE colonization at some point during admission. The most prevalent gene was *bla*_NDM_, but *bla*_OXA-48_ and *bla*_VIM_ were also identified [22].

**Libya:** In 2011, a case of OXA-48 *K. pneumoniae* rectal carriage was reported in a patient transferred from Libya to Slovenia [83], and in 2012–2013 other *K. pneumoniae* and *E. coli* encoding for OXA-48 and NDM-1 enzymes were isolated from Libyan patients [84,85,86]. Later, in 2016, more such strains isolated in Libya and Tunisia were described, with 11.4% of all studied strains being *K. pneumoniae* OXA-48 producers [87].

**Madagascar**: An article from 2015 reported community colonization with NDM-1 *K. pneumoniae* (0.3% CPE intestinal carriage) [88]. A 2020 study reported six cases of CPE originating from Madagascar, isolated between 2011 and 2016, and an increasing CPE fecal carriage prevalence in all recruited countries (Madagascar, French Reunion, Mauritius, Seychelles, India and Mayotte/Comoros) [89].

**Mauritius**: In 2012, an MDR strain of *K. pneumoniae* isolated from a patient in Mauritius in 2009 was reported to be *bla*_NDM-1_ positive [90]. A 2020 study reported 11 more cases of CPE originating from Mauritius, isolated between 2011 and 2016 [89].

**Malawi:** A study published in 2019 reported that 16 out of 200 (8%) Enterobacterales isolated in 2016–2017 in Malawi were *Klebsiella* spp. and *E. coli* producing KPC-2, NDM-5 and OXA-48 enzymes [91]. 

**Mali:** In 2017, an article reported an OXA-181 *E. coli* among 82 *Enterobacteriaceae* isolated from 1334 positive blood cultures, probably the first CPE reported in Mali [92]. In 2023, a study was published that evaluated 526 patients with pleurisy between 2021 and 2022. One hundred and ten were diagnosed with enterobacterial pleuritis, mainly *E. coli, K. pneumoniae* and *Proteus mirabilis*. Three isolates (2.72%), one *K. pneumoniae* and two *Providencia* spp., tested positive for *bla*_NDM-1_ [93].

**Morocco:** In 2011, the emergence of NDM-1-producing *K. pneumoniae* was reported in Morocco [94]. In a study published in 2012, in which 463 Enterobacterales isolated in 2009–2010 were evaluated, 2.8% were CPE: OXA-48 or NDM-1, *Klebsiella* spp. or *Enterobacter cloacae* [95]. Later, more CPE were reported: OXA-48 and IMP-1 *E. coli* (2/1174, 0.17%), in 2013 [96]; OXA-48 and NDM-1 *K. pneumoniae* (11/166, 6.62%), in 2015 [97]. A 2014 published study reported that, in 2012, among 77 patients screened by rectal swabbing and culture on screening media followed by PCR, 10 OXA-48 CPE intestinal carriers (13%) were found. The prevalent species were *K. pneumoniae* and *E. cloacae* [98]. A 2017 study reported 3 CPE *bla*_OXA-48_ among 169 *Enterobacteriaceae* isolates from 164 neonates evaluated for ESBL and CPE rectal carriage (1.8% CPE carriage) [99]. In a 2021 published study, it was reported that 641 *Enterobacteriaceae* were isolated from 455 newborns and infants screened for intestinal colonization on admission (2013–2015). A total of 8.7% were colonized with *bla*_OXA-48_ CPE. During hospitalization, 207 newborns were included in a follow-up acquisition study, and it was observed that 12.5% had acquired *bla*_OXA-48_ CPE during their hospital stay. The majority of CPE consisted of *K. pneumoniae* and *E. coli* [100]. A 2022 study in which GNB isolated in 2018–2020 were analyzed reported that out of 810 Enterobacterales, 210 were eligible for β-lactamase screening: 40 presented NDM and 39 presented OXA enzymes; 7 carried both OXA-48 and NDM-1. These findings indicate a CPE prevalence of ~10% [101]. A study from 2023 which evaluated 195 CRE isolated from 18,172 clinical samples identified 190 CPE (~1%), of which 74 were biofilm-associated MBL producers. Sixty-two of seventy-four (62/74) presented *bla*_NDM_, and *bla*_NDM_ and *bla*_OXA-48_ were found to be associated in twelve strains. *K. pneumoniae* was the most prevalent species [102]. Another 2023 study evaluated 199 positive NICU blood cultures from 2019. Seventy-five of one hundred and ninety-nine (75/199) were Enterobacterales, and thirty-six out of seventy-five were CPE (especially *K. pneumoniae* and *Enterobacter* spp. encoding for OXA-48 and/or NDM). Thus, CPE were responsible for 18% of 199 positive blood cultures [103]. One more 2023 study, which included 38 MDR Enterobacterales, especially *E. coli*, *Klebsiella* spp. and *Enterobacter* spp., isolated in 2016–2017 from clinical samples, identified 22 CPE positive for *bla*_OXA-48_ and *bla*_NDM_ [104]. The overall CPE colonizations in Morocco varied from 1% to 13%, but higher percentages were observed in symptomatic infections.

**Mozambique:** A 2021 published study reported the emergence of *E. coli bla*_NDM-5_ [105].

**Namibia:** In a study published in 2022, among 13,673 positive urine cultures from 2016–2017, resistance to carbapenems was low and only one CPE was found [106].

**Nigeria:** In 2013–2014 reports, several strains isolated in Nigeria and evaluated with phenotypic assays were CR and suspected as CP (*n* = 9 of 97 tested strains [107]) or confirmed as CP (*n* = 10 of 182 tested strains [108]). In 2015, a rectal swab was collected from a patient previously hospitalized in Nigeria, and the patient was found to be colonized with NDM-1 *K. pneumoniae*, OXA-181 *E. coli* and VIM-2 *P. aeruginosa* [109]. In 2017, among 248 evaluated clinical isolates (140 *E. coli* and 108 *K. pneumoniae*), 191/248 were identified as CR and 93/191 (41 *E. coli* and 52 *K. pneumoniae*) were identified as CPE by MHT. An increase in CPE prevalence (from 11.9% to 39.2%) was observed when the results were compared to 2011 reports [110]. In 2019, an outbreak of five NDM-5-producing *Klebsiella quasipneumoniae* was reported [111]. Later, in a 2020 study, 397 Gram-negative bacterial strains (of which 293 were Enterobacterales) isolated from patients were tested. Fifteen of three hundred and ninety-seven (15/397) GNB (7/293 Enterobacterales, 2.38%) were Carba NP positive [112]. In a 2021 study, out of a total of 134 *K. pneumoniae* strains isolated in three Nigerian hospitals, 11 (8.2%) were CPE: 8 presented *bla*_NDM-1_, 2 presented *bla*_NDM-5_ and 1 presented *bla*_OXA-48_ [113]. A 2022 study on 107 *E. coli* clinical isolates revealed that 6 (5.6%) presented *bla*_NDM-1_ and *bla*_NDM-5_ [114]. In 2023, 33/49 strains of MDR Enterobacterales were identified as CPE with *bla*_NDM_ and *bla*_OXA-48-like_ gene associations. It was observed that three strains were susceptible to meropenem [115]. In a 2023 study that also included Kenya, 20/24 Nigerian infants presented CPE colonization at some point during hospital admission. *bla*_NDM_ was identified especially, but *bla*_OXA-48_ and *bla*_VIM_ were also identified [22]. More noteworthy recent data were found in a study on Sub-Saharan countries described below [116]. It is still difficult to draw a general conclusion regarding CPE colonization in Nigeria, as data strictly regarding this topic are scarce. However, the overall CPE prevalence ranged from 2.38 to 39.2% or more.

**São Tomé and Príncipe:** In a 2018 study, it was reported that out of 50 patients screened for MDR-GNB presence, 34 CRE were isolated from 22 patients. The 34 strains were *E. coli* and *K. pneumoniae*, which harbored *bla*_OXA-181_, resulting in 44% CR CPE colonization [117].

**Senegal:** In 2011, eight *K. pneumoniae* strains and one *E. coli* strain isolated from Senegalese patients during 2008–2009 were PCR-confirmed to have the *bla*_OXA-48_ gene. As imipenem (and meropenem) were susceptible, such strains could pass undetected and the importance of routine AST was raised [118].

**Sierra Leone:** A 2013 study recorded strains of *K. pneumoniae*, *E. coli* and *E. cloacae* presenting *bla*_OXA-51_ and *bla*_OXA-58_—genes usually found in *Acinetobacter* spp.—among 20 GNB isolated between 2010 and 2011 in a Sierra Leone hospital [119]. 

**South Africa:** In 2011, the first reports of NDM-1 and KPC-2 *K. pneumoniae* isolated from patients in South Africa, along with the first case of KPC in Africa, were published [120]. In 2013, a paper was published describing the emergence of OXA-48-like (including OXA-181)-producing *K. pneumoniae* in hospitalized patients (2011–2012). One patient who previously received a kidney transplant in Egypt was probably the first case of OXA-48 reported in South Africa [121]. Later, an article from 2019 characterized several OXA-48-like CPE, including OXA-181 [122], while another article reported an increase in CRE prevalence from 2.6% (2013) to 8.9% (2015) in an NICU. A total of 22/26 CRE were *K. pneumoniae*, and 17/18 tested CRE presented NDM or VIM enzymes [123]. A study published in 2019, in which 439 patient samples (438 rectal swabs and 1 stool sample) collected in 2016 were screened for intestinal colonization, identified 12 CRE but only 1 *K. pneumoniae* harboring *bla*_NDM-1_ (0.22%) [124]. In one of the 2020 studies, 5/263 (1.9%) rectal swabs and 5 other isolates from infected patients were confirmed as CR *K. pneumoniae*. All 10 isolates showed genotypic resistance, being *bla*_NDM-1_ positive. Sequencing revealed genetic relatedness, with the same plasmid multilocus sequence type and capsular serotype, thus supporting the horizontal transfer of resistance genes and clonal dissemination [17]. Another study evaluated ESBL and CRE rectal colonization in a pediatric hospital. Although 1/200 patients presented CR *E. cloacae* colonization, no common CP gene was found [125]. Other 2020 studies reported more OXA-48 and NDM *K. pneumoniae* strains isolated from clinical samples, such as blood cultures; similar strains were identified in carriers [126,127]. A study from 2021 which screened 31 ICU patients by collecting 97 rectal swabs which were cultivated on screening media isolated 14 CR *K. pneumoniae*, and all were confirmed as CPE through molecular testing (all harboring *bla*_OXA-181_) [128]. In a 2022 screening article, out of 587 samples collected from humans (230 rectal swabs), pigs (345 rectal swabs) and water (12), 19 (3.2% of total) presented CRE, of which 9 presented *K. pneumoniae*. Of the 19 samples, 4 were environmental and 15 were human in origin (resulting in 6.5% colonized humans). Sixteen of nineteen (16/19) also tested positive for OXA-181 (9/16) and NDM-1 (4/16), but OXA-48, GES-5 and OXA-484 were also identified [129]. A 2022 publication of a large 2019–2020 surveillance study reported 2144 patients with CRE bacteremia from multiple healthcare facilities. Out of 1082 studied strains, 863 (79.8%) were *K. pneumoniae*, followed by *E. cloacae, S. marcescens* and *E. coli* in close proportions. A total of 915/1082 (84.6%) presented one carbapenemase gene, while 38 (3.5%) had two genes encoding for carbapenemases. The most common carbapenemase gene was *bla*_OXA-48-like_ (761/991, 76.8%), followed by *bla*_NDM_ (209/991, 21.1%), *bla*_VIM_*, bla*_GES_ and *bla*_KPC_ [130]. In a 2023 study, 23/53 newborns that suffered infections in a neonatal unit had CRE-positive cultures, and 15/33 newborns screened for CRE carriage by rectal swabs tested positive. For 20 of the strains, *bla*_NDM_ and *bla*_OXA-48_ genes were identified [131]. Another 2023 study revealed *bla*_OXA-48-like_ genes in 18/39 CR *Serratia marcescens* isolated from patients during 2015–2020. It must be noted that a total of 1396 *S. marcescens* strains were identified, and only 21 of the 39 CR were also sequenced. In total, 19 of the 21 patients were on antibiotics prior to isolation [132]. More noteworthy recent data were found in a recent study on Sub-Saharan countries described below [116]. Overall, CPE colonization in South Africa was found to range from 0 to 6.5% or more (close to 50% if studies on small lots are taken into account). 

**Somalia:** Although data are very limited for Somalia, in a 2021 study that evaluated carbapenemase-encoding bacteria (CEB) isolated between 2014 and 2019, 11 German residents of Somali descent tested positive for CEB, with genes encoding for NDM, OXA-23 and VIM [133].

**Sudan:** In a 2018 study, 36.1% of 200 Gram-negative strains isolated in Sudan were found to be MBL producers [134], while a 2020 study identified an important number of *K. pneumoniae* strains (*n* = 46) harboring genes encoding for OXA-48, NDM, KPC and IMP that were isolated from infected patients [135]. A study from 2021 reported that, out of 206 CR-GNB, 171 where phenotypically confirmed as CR and 121 harbored carbapenemase genes (including CPE, mostly *K. pneumoniae* and *E. coli*), such as *bla*_NDM_ (107), *bla*_IMP_ (7), *bla*_OXA-48_ (5) and *bla*_VIM_ (2), with 3 strains co-harboring *bla*_NDM_ and *bla*_OXA-48_, 1 strain co-harboring *bla*_NDM_ + *bla*_VIM_ and 1 strain co-harboring *bla*_NDM_ + *bla*_IMP_ [136]. In a 2023 article, 86 *K. pneumoniae* hospital isolates (2016–2020) were evaluated. In total, 35 were CR, and 3/35 were not CPE. However, the study indicated that among the total 86 sequenced strains, 37 were CPE, encoding for NDM-1, NDM-4, NDM-5, OXA-48 and OXA-232; 3 strains presented both NDM-5 and OXA-48 [137].

**Tanzania:** A study from 2014 showed that, in Tanzania, 80 of 227 (35.24%) MDR-GNB (among which 176 were Enterobacterales) presented one or more genes encoding for carbapenemases: IMP, VIM, OXA-48, KPC and NDM [138]. In a 2020 study, 244 *Enterobacteriaceae* were isolated from 194/595 HIV-positive patients screened by collecting rectal swabs. For one patient, rectal colonization with CP *E. coli* was reported (0.16% CPE fecal carriage among all participants; 0.5% CPE fecal carriage among participants with positive cultures) [139]. In 2023, a study reported a CPO isolation rate of 22.8% from hospital surfaces [140].

**Tunisia:** In 2010, the first warnings were released on OXA-48 *K. pneumoniae* in Tunisia [141]. In 2011, an outbreak of OXA-48 *K. pneumoniae* was reported, with 21 out of 153 CR strains testing positive for this enzyme [142], followed by other reports of OXA-48 *K. pneumoniae* and *Citrobacter freundii* in 2012 [143] and the case of a Libyan patient infected with a *K. pneumoniae* co-harboring NDM-1 and OXA-48 in 2013 [86]. In 2015, two patients who underwent rectal swab screening in 2015 after being transferred from Tunisia to Poland presented with *bla*_NDM-1_
*K. pneumoniae* and *bla*_OXA-48_
*K. pneumoniae* colonization. Ten days after admission, *bla*_NDM-1_
*K. pneumoniae* and *E. coli* were found in one patient, with a gene similar to the one isolated in the other patient [144]. Later papers reported KPC-2 *E. coli* and OXA-48 and VEB-8 *K. pneumoniae* (2016–2017) [145,146]. A large study from 2019 phenotypically tested 2160 *K. pneumoniae* strains and reported 342 CR strains (15.8%), 203 being suspected of OXA-48-like enzymes and 17 of MBL (10% of *K. pneumoniae* strains were CP) [147]. Another 2019 study evaluated intestinal MDR-GNB carriage in 38 patients at admission and then weekly. During their stay, 14 of them were colonized with various MDR-GNB, among which 10 CR-GNB were identified. Among *Enterobacteriaceae*, five CPE (four OXA-48 and one NDM) were identified [148]. A study from 2021 which characterized 19 *Klebsiella oxytoca* strains isolated in a Tunisian hospital (2013–2016) showed that all these strains presented the *bla*_OXA-48_ gene [149]. In a 2022 study, out of 2135 stool samples collected from food handlers between 2012 and 2017, 7 (0.33%) were positive for CPE carriage (OXA-48 and NDM-1 *K. pneumoniae* and *E. coli*) [150]. Similar strains were described by other authors in 2022 [151]. Another 2022 study, in which 227 hospitalized children were screened for MDR *Enterobacteriaceae* rectal colonization, reported only 1 patient (0.44%) with CPE carriage (a strain of *bla*_OXA-48_
*Klebsiella oxytoca*) [152]. In 2023, the first report of IMI-2-producing *Enterobacter bugandensis* isolated from the stool of a healthy volunteer in Tunisia was published [153]. Overall, although significant rates of CPE were observed generally, CPE carriage seems to be under 1% in Tunisia.

**Uganda:** In a 2015 study, it was reported that 56 of 658 (8.5%) Enterobacterales strains (especially *K. pneumoniae* and *E. coli*) isolated in 2013–2014 from a Ugandan hospital encoded for carbapenemases (confirmed by RT-PCR). Eleven of these fifty-six strains encoded for VIM and OXA-48 enzymes and presented phenotypically detectable resistance [154]. In a 2020 study, 15 of 69 GNB isolated from surgical site infections and identified as *K. pneumoniae* were suspected as CPE [155]. Later, in 2021, in a study where 227 virulent *K. pneumoniae* strains isolated from four hospitals in 2019 were evaluated, it was shown that 23.3% of the strains were phenotypically CR, but the PCR analysis revealed that even more (43.1%) presented genes associated with CP, especially *bla*_OXA-48-like_, *bla*_IMP_, *bla*_VIM_, *bla*_KPC_ and *bla*_NDM_ [156]. In a 2023 study, 95/192 (49.5%) *E. coli* strains isolated from stool samples collected in equal amounts from humans (49/96) and their livestock (45/96) presented *bla*_KPC_ on PCR evaluation, although not all were phenotypically resistant to carbapenems and not all CRE were CPE [157]. In another 2023 study, multiple samples (swabs) were collected from 137 mothers and their 137 newborns, 67 health workers, and 70 frequently touched hospital surfaces. One hundred and thirty-one (131) GNB were isolated from 21 mothers, 15 babies, 2 health workers and 13 surfaces, of which 104/131 were *K. pneumoniae*, *E. coli* and *Enterobacter* spp. In total, 10/104 strains were CR, 6/10 were confirmed as CPE by PCR (*bla*_VIM_, *bla*_IMP_ and *bla*_NDM_) and 4/6 co-harbored more than one carbapenemase gene. The overall CPE prevalence was 1.46% in this study [158]. The difference between results regarding CPE colonization is significant: 1.46% for one study (maybe less if surfaces were excluded) and 49.5% for another study. More studies are necessary in order to draw a conclusion.

**Central Africa:** A systematic review from 2023 evaluated all publications from 2005 to 2020, including Gabon, Cameroon, the Democratic Republic of Congo, the Central African Republic, Chad, the Republic of Congo, São Tomé and Príncipe, and Angola. The revealed data regarding CPE were still scarce for these countries but nonetheless relevant. In Angola, NDM-1-, NDM-5- and OXA-181-producing strains were found in clinical and intestinal carriage human isolates, and CPE isolation rates were in the range of 26.4–78% (these data are similar to those contained in the reports presented above). In Cameroon, NDM-1 and NDM-4 were described, and it should be noted that *bla*_AIM-1_ was identified in the environment, among other genes; in Chad, NDM-5 and OXA-181 were observed, with 2.5–6.5% CPE; in Gabon, NDM-7 and OXA-48 were observed, with 5.1% CPE (these findings being close to those of a study presented above); in the Republic of Congo, OXA-181 was observed, with 6.97% CPE; and in São Tomé and Príncipe, OXA-181 was observed, with 44% CPE (the study was described above). For the Democratic Republic of Congo, OXA-48-, KPC-, VIM-, IMP- and NDM-encoding genes were found in wastewater and drinking water. No data were available for the other included countries [159].

**Sub-Saharan Africa:** A study from 2023 evaluated data on MDR-GNB from Cameroon, Ivory Coast, Nigeria and South Africa. In total, 5014 GNB isolates were included, of which 3905 were Enterobacterales, 214 of which were CRE. *K. pneumoniae* was the most prevalent CRE (72.4%). Of the Enterobacterales that underwent molecular characterization, 136 (3.5% of all Enterobacterales) carried an MBL (131 were NDM, all were CR and 5 were VIM). Most NDM strains were from Nigeria (87/512 characterized strains, 17%), followed by Cameroon (5/42, 11.9%), South Africa (37/444, 8.3%) and Ivory Coast (2/56, 3.6%). The 5 VIM isolates were from South Africa, while 25 NDM strains also carried *bla*_OXA-48_-like genes. Out of the 127 strains that were non-MBL CPE (3.3% of all Enterobacterales), 125 were OXA-48 group carriers (105 carried OXA-181, 15 carried OXA-48 and 5 carried OXA-232) and 2 were KPC carriers. Including the 25 OXA-48 + MBL strains, the OXA-48/OXA-48-like isolates were most prevalent in South Africa (129/444 molecularly characterized strains, 29.1%), followed by Cameroon (5/42, 11.9%), Nigeria (15/512, 2.9%) and Ivory Coast (1/58, 1.8%). The two KPC strains were from South Africa [116]. 

In some regions, OXA-48 and VIM-2 *Salmonella enterica* ser. Kentucky were reported, according to a study published in 2013 [160]. 

Also, reports of CP and CR *Acinetobacter* spp. and *Pseudomonas* spp. increased in number, with alarmingly high rates of resistance [161,162,163,164,165]. Even rare species of CP non-fermenters were reported [166].

A summarized version of the identified genes responsible for carbapenemase production in Enterobacterales is presented in Table 1. A comparison between geographical regions identified a number of differences concerning the current burden.

In the case of Ambler class A encoding genes, the *bla*_KPC_ gene in at least one variant was identified in all presented regions, while others, such as *bla*_GES_ or *bla*_IMI_, were identified only in southern and western Africa and in northern Africa, respectively.

The genes encoding the production of metallo β-lactamases followed a similar pattern, with *bla*_NDM_ and *bla*_VIM_ variants being present all across Africa, while others, such as *bla*_IMP_, were not identified in southern or the Sub-Saharan regions. The emergence of other resistance genes, such as *bla*_AIM-1_ in Central Africa, represents a concerning evolution in regard to the global spread of AMR and should be thoroughly investigated in order to contain further spread.

The most important variability in resistance determinants was observed for the Ambler class D enzymes. While *bla*_OXA-48_ was found in all geographic regions, other oxacillinases were identified only in specific parts of Africa. Some genes were only reported regionally: *bla*_OXA-204_ in northern Africa, *bla*_OXA-58-like_ genes alongside others (*bla*_OXA-241_ and *bla*_OXA-244_) in western Africa and *bla*_OXA-484_ in southern Africa. However, *bla*_OXA-181_ was detected in all regions except northern Africa, while *bla*_OXA-232_ was not identified in Central Africa. One particular characteristic was the presence in Enterobacterales of genes naturally found in microorganisms pertaining to other genera. Genes such as *bla*_OXA-51-like_ (64, 65, 71, etc.) [119] and *bla*_OXA-484_ [129], which can be naturally found in *Acinetobacter* spp., are rare findings in association with Enterobacterales and are now reported through sequencing. These results raise serious public health issues that need to be addressed in order to stop the further spread of resistance mechanisms.

Further details of the studies included in our research can be found in Appendix A.

## 3. Discussion

Antimicrobial resistance is a global public health challenge causing high morbidity and mortality. CPE-related infection and carriage have been reported worldwide. The high mobility of people who may be asymptomatic carriers in and out of Africa contributes to the ease of spread of CPE genes. The burden of CPE includes the clinical impact but also the influence on the healthcare system, as prolonged treatment and hospital stays are associated with higher costs. 

Globally, most reports of CPE come from Europe, Asia and North America [167]. To our knowledge, this review presents the latest updated information on CPE in the African continent since the review published by Manenzhe et al. in 2014 [168]. This study could be of particular interest to researchers who want to undertake systematic evaluations of carbapenemase production in Enterobacterales and as a starting point for evaluating current published data as well as the methodological drawbacks of the studies published thus far.

Although not all regions are well represented, this review showed a great diversity of carbapenemase genes from all Ambler classes throughout the African continent. Of particular interest is that, in some cases, strains containing multiple carbapenemase-encoding genes were identified.

It is hard to say with certainty when, where or how CPE began to spread in Africa, as there are many factors involved, but the first studies describing the emergence of CPE evaluated strains isolated in the mid- or early 2000s. These strains were disseminated in various ways, including asymptomatic carriers. It should be noted that some studies reported data from the same year the study was published, while others reported data from previous years.

Limiting factors of the studies considered include the following: the published data related only to MDROs and not all isolated bacteria; the reported findings were for all Gram-negative bacteria and did not separate Enterobacterales from the other bacteria identified; only some of the isolates belonging to one species were analyzed; there was a lack of confirmatory tests (such as sequencing); sequencing was performed for particular genes not looked for or identified in Enterobacterales; confirmatory tests were only available for a limited number of strains or genes; and only a limited number of strains was included. These factors are important, as particular, rare findings were reported in some regions. Also, situations that render the data inconclusive were observed: some studies reported OXA-48-like enzymes or *bla*_OXA-48_-like genes as OXA-48 or *bla*_OXA-48_ in the absence of techniques that offer certain results (generally, sequencing). Other enzymes and genes, such as GES and *bla*_GES_, were reported without specifying the exact subtype. This is important, as not all OXA-48-like and GES enzymes are carbapenemases (and not all *bla*_OXA-48-like_ and *bla*_GES_ genes encode for carbapenemases). Also, there is some uncertainty associated with the fact that a few studies reported bacterial isolates that presented three or even four carbapenemase encoding genes. 

There are important differences between regions and countries and sometimes between healthcare units, screened population groups and/or evaluated clinical samples. Although CPE are more frequently associated with infections, colonization may occur in asymptomatic individuals, both hospitalized and from communities. The most common CPE species involved in infections and colonization seem to be *K. pneumoniae* and *E. coli*, but *Enterobacter* spp. are also frequent, while the most prevalent associated carbapenemases are NDM, OXA-48/OXA-48-like and, to some extent, VIM and KPC enzymes—results that match the existing literature [169,170,171,172,173,174].

In the past 5–7 years, reports of highly resistant CPE have become increasingly common, and CPE which associate multiple resistance mechanisms, including carbapenem and colistin resistance (e.g., *mcr*-1) or multiple carbapenemases, have emerged in Tunisia, in 2017 [175]; Egypt, in 2021 [176]; Sudan, in 2021 [136]; Ethiopia, in 2021 [64]; Uganda, in 2021 [156]; Ghana, in 2023 [79]; and again in Sudan, in 2023 [137]. A 2019 report detailed a patient who was recently admitted to a Kenyan hospital and tested positive for both *Candida auris* and CPOs [177]. These may have been caused by the long-term rise in CPE prevalence and the rise in carbapenem prescriptions, which favors the selection of resistant strains; the rise in the accessibility of certain testing techniques, including phenotypic and molecular testing, should also be considered [178]. Unfortunately, although carbapenems are already expensive and still difficult to access for the population in some countries, antimicrobial molecules active against CPE will be necessary in Africa [116,117,179].

Some studies have shown that molecular analysis might reveal even more carbapenemase-producing (or carbapenemase-encoding) Enterobacterales than phenotypic tests among strains with no expressed CR, which supports the concern that some CPE can be missed by usual screening methods and could disseminate silently [137,156]. Other studies revealed quite the opposite, showing that not all CROs are CP and that carbapenem resistance can occur through other mechanisms—facts supported by EUCAST and different studies [27,79,136]. This aspect may be dependent on the type of carbapenemase, the species and the virulence of the strain in question [116].

Scarce or no published data regarding CR or CPE were found for some regions, especially for developing countries or countries where access to carbapenems is limited [180,181]. For many countries, reported data on CPE carriage were inconclusive. As this study is not a systematic review, some reported data might have been omitted.

A method of surveillance for carriers with MDROs that could be accessible even for countries with limited resources is the use of screening culture media. Relevant clinical samples (e.g., rectal swabs or fecal matter for CPE, tegumentary swabs for MRSA, etc.) can be collected periodically or at certain times (at the moment of admission to a hospital, during a hospital stay, before surgery, before release, on transfer to another healthcare facility, etc.) and cultivated on selective and differential media specially designed for the identification of certain microorganisms. This method is easy to use and has proven effective, as some studies show [23,45,98,128]. However, further phenotypic or molecular assays are necessary to confirm carbapenemase production in the isolates that grow on the screening media [23,27,182,183].

In addition to human colonization, MDROs (including CPE) can also be spread by contaminated surfaces; hands [184]; money [185]; contaminated food [4,186,187]; soil, water and air [5,188,189,190,191]; colonized animals [39]; birds [192], including migratory species [193]; and even insects (e.g., cockroaches and flies) [194,195,196].

There are certain organizations around the world that contribute to the fight against the spread of AMR. One such example is the Pasteur Network [197], which is already present in some parts of Africa (e.g., Cameroon, Niger, Côte d’Ivoire, Madagascar, etc.) and other parts of the world (the Americas, the Asia–Pacific and the Euro-Mediterranean). Further collaborations between public health institutions and the Pasteur Network, as well as other networks worldwide, will allow experts from around the world to come together and focus on addressing difficult issues, including AMR. These partnership prospects are welcomed by international and national committees, including well-established representative institutions. The Cantacuzino National Military Medical Institute for Research and Development and the Carol Davila University of Medicine and Pharmacy are willing to be involved in such collaborations. 

Several other limitations regarding this review were identified:Although significant progress has been made towards implementing efficient measures of surveillance and control, follow-up data are still lacking. When available, the scarcity and heterogeneity of studies hinders the prospect of researchers being able to properly analyze data. Due to this limitation, it was not always possible to specify data important for this review (e.g., numbers of CRE and CPE, genetic profiles regarding carbapenemase-encoding genes, etc.), as they were uncertain, not available or not applicable (N/A);As English-language publications were mainly accessed in order to write this review, studies presenting relevant information published in French or other languages might have been overlooked;Although extensive research was performed in order to extract the information, studies that did not match the search criteria and keywords but which could have contained important data might have not been identified;It should be mentioned that this study did not extensively analyze data for other CP Gram-negative bacteria, such as *Pseudomonas* and *Acinetobacter*, that may present with different enzymes and a different epidemiology.

## 4. Materials and Methods

The data included in this non-systematic review were revised in October 2023. In order to extract the information for this study, two database queries were employed. To conduct the aforementioned searches, the Medical Subject Headings (MeSH) technique was used, while parallel strategies employing identical keywords were used for the other available databases.

The first search was conducted in PubMed, PubMed Central (PMC) and Google Scholar. The keywords “carbapenemase”, “colonization” and “Africa” were used both alone and with Boolean operators to narrow down the results. 

The second search was conducted in Web of Science, PubMed, Cochrane, ScienceDirect, PubMed Central and Google Scholar. The research papers were extracted using keywords such as “carbapenem-resistant”, “carbapenemase”, “beta-lactamase”, “Enterobacterales”, “*Enterobacteriaceae*”, “Gram-negative”, “colonization”, “screening”, “travelers”, and “Africa” or African region and country names. The keywords were used both alone and with Boolean operators to narrow down the results.

Original articles were prioritized over reviews. Systematic reviews or literature reviews that offered insights regarding the epidemiology of CRE and/or CPE in the African continent were included. Several studies that approached antimicrobial resistance in non-fermentative Gram-negative bacilli were included to offer an epidemiological viewpoint on the topic; however, they were not extensively discussed.

The inclusion criteria used for our research were defined as research papers presenting reports and data on carbapenemase-producing and/or carbapenem-resistant Enterobacterales and colonization with such microorganisms in countries and regions located in Africa. Articles describing cases of African-origin patients or people who were hospitalized or traveled in Africa were also included if the data were relevant.

The exclusion criteria included research articles investigating other Gram-negative bacilli, those that focused on mechanisms of resistance that affected susceptibility to carbapenems other than carbapenemases, articles that did not include data in regard to CPE and/or CRE, works with unclear methodologies in regard to the processing of clinical samples, papers that did not discuss African countries and studies not written in English.

For reference management, Zotero version 6.0.27 was used.

## 5. Conclusions

Despite some limitations concerning the availability of data, this study presents the most recent information about CPE in Africa. 

Although the exact origin of CPE in Africa is impossible to pinpoint, CPE are tracked back to the early to mid-2000s. After the first CPE genes emerged, they started to spread locally and regionally, sometimes encoding multiple carbapenemases belonging to different Ambler classes but also affecting susceptibility among multiple antibiotic classes. The burden of this phenomenon is both clinical and economical, as the concomitant presence of multiple carbapenemases can severely reduce the efficacy of treatment while being associated with higher costs due to longer hospitalization periods.

The increase in the number of studies published in recent years that describe the detection of CPE in Africa may be attributed to the greater accessibility of testing. However, the methods needed to properly identify and characterize strains are yet to be made generally available, and this stands as a possible reason for why not all countries have reported data. 

The general context of the current spread of genes encoding carbapenemase production can now be outlined in the African continent. However, because of the heterogeneity of the available data and the scarcity of data in some regions, additional research is required. The results of our study offer an up-to-date perspective on the topic while also underlining the need for further studies.

Interdisciplinary collaboration between microbiologists, epidemiologists and clinical infectionists is essential to limit the spread and reduce the overall burden of CPE carriage and infection.

## Figures and Tables

**Figure 1 antibiotics-13-00295-f001:**
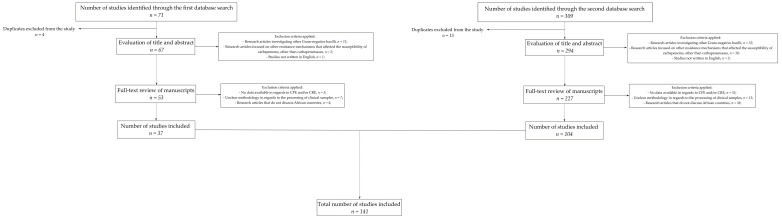
Flowchart of the selection process for the data included in the study.

**Table 1 antibiotics-13-00295-t001:** Genes encoding for carbapenemase production in Enterobacterales reported in Africa by geographical region. PT, phenotypic test; PCR, polymerase chain reaction; Seq, sequencing methods; PFGE, pulsed-field gel electrophoresis; Lat. fl., lateral flow immunochromatographic test; SCM, screening culture media; SChM, selective chromogenic media for CP-GNB; SM, selective media.

African Region	Country	No. of Studies per Country	Genes Identified	Genes Identified/Region	Detection Method for CP	Countries Not Represented in Table
**Northern Africa**	**Algeria**	8	*bla*_NDM-5_, *bla*_VIM-19_, *bla*_OXA-48_	*bla* _IMI-2_ *bla* _KPC_ *bla* _KPC-2_ *bla* _IMP_ *bla* _IMP-1_ *bla* _IMP-4_ *bla* _NDM_ *bla* _NDM-1_ *bla* _NDM-4_	*bla* _NDM-5_ *bla* _NDM-6_ *bla* _NDM-7_ *bla* _VIM_ *bla* _VIM-1_ *bla* _VIM-2_ *bla* _VIM-19_ *bla* _OXA-48_ *bla* _OXA-204_ *bla* _OXA-232_	PT, PCR, Seq	Canary Islands Madeira Islands
**Egypt**	6	*bla*_KPC_, *bla*_IMP_, *bla*_NDM_, *bla*_NDM-1_, *bla*_VIM_, *bla*_VIM-2_, *bla*_IMP-4_, *bla*_OXA-48_	PT, PCR, PFGE, Seq
**Libya**	5	*bla*_NDM-1_, *bla*_OXA-48_	PT, PCR, Seq
**Morocco**	11	*bla*_IMP-1_, *bla*_NDM_, *bla*_NDM-1_, *bla*_NDM-7_, *bla*_OXA-48_	SCM, PT, Lat. fl., PCR, Seq
**Sudan**	4	*bla*_KPC_, *bla*_VIM_, *bla*_IMP_, *bla*_NDM_, *bla*_NDM-1_, *bla*_NDM-4_, *bla*_NDM-5_, *bla*_NDM-6_, *bla*_OXA-48_, *bla*_OXA-232_	PT, PCR, Seq
**Tunisia**	14	*bla*_IMI-2_, *bla*_KPC-2_, *bla*_NDM-1_, *bla*_VIM-1_, *bla*_OXA-48_, *bla*_OXA-204_	PT, PCR, PFGE, Seq
**Eastern Africa**	**Djibouti**	1	*bla*_NDM-1_, *bla*_NDM-5_, *bla*_OXA-48_, *bla*_OXA-181_	*bla* _KPC_ *bla* _KPC-2_ *bla* _IMP_ *bla* _IMP-10_ *bla* _NDM_ *bla* _NDM-1_ *bla* _NDM-4_ *bla* _NDM-5_ *bla* _NDM-7_	*bla* _VIM_ *bla* _OXA-23_ *bla* _OXA-48-like_ *bla* _OXA-48_ *bla* _OXA-181_ *bla* _OXA-232_	SChM, PT, Lat. fl., Seq	BurundiComorosEritreaReunionRwandaSeychellesSouth SudanZambiaZimbabwe
**Ethiopia**	14	*bla*_KPC_, *bla*_NDM_, *bla*_NDM-1_, *bla*_NDM-5_, *bla*_OXA-181_	SChM, PT, PCR, PFGE, Seq
**Kenya**	5	*bla*_NDM_, *bla*_NDM-1_, *bla*_NDM-5_, *bla*_NDM-7_, *bla*_VIM_, *bla*_OXA-48_, *bla*_OXA-181_, *bla*_OXA-232_	PT, PCR, PFGE, Seq
**Madagascar**	2	*bla*_NDM-1_, *bla*_NDM-4_, *bla*_NDM-5_	PT, PCR, PFGE, Seq
**Malawi**	1	*bla*_KPC-2_, *bla*_NDM-5_, *bla*_OXA-48_	SCM, PT
**Mauritius**	2	*bla*_IMP-10_, *bla*_NDM-1_, *bla*_NDM-5_, *bla*_NDM-7_, *bla*_OXA-181_	PT, PCR, PFGE, Seq
**Mozambique**	1	*bla* _NDM-5_	Seq
**Somalia**	1	*bla*_NDM_, *bla*_OXA-23_, *bla*_VIM_	Seq
**Tanzania**	3	*bla*_KPC_, *bla*_NDM_, *bla*_IMP_, *bla*_VIM_, *bla*_OXA-48_	SCM, PCR
**Uganda**	5	*bla*_KPC_, *bla*_IMP_, *bla*_NDM_, *bla*_VIM_, *bla*_OXA-48-like_	PCR
**Southern Africa**	**Botswana**	1	-	*bla* _GES-5_ *bla* _KPC_ *bla* _KPC-2_ *bla* _KPC-3_ *bla* _NDM_ *bla* _NDM-1_ *bla* _NDM-2_	*bla* _NDM-7_ *bla* _VIM_ *bla* _OXA-48-like_ *bla* _OXA-48_ *bla* _OXA-181_ *bla* _OXA-232_ *bla* _OXA-484_	SChM, Unidentified genotypic method	LesothoSwaziland
**Namibia**	1	-	-
**South Africa**	15	*bla*_GES-5_, *bla*_KPC_, *bla*_KPC-2_, *bla*_KPC-3_, *bla*_NDM_, *bla*_NDM-1_, *bla*_NDM-2_, *bla*_NDM-7_, *bla*_VIM_, *bla*_OXA-48-like_, *bla*_OXA-48_, *bla*_OXA-181_, *bla*_OXA-232_, *bla*_OXA-484_	SCM, PCR, PFGM, SEQ
**Central Africa**	**Angola**	2	*bla*_NDM-1_, *bla*_NDM-5_, *bla*_OXA-181_	*bla* _KPC_ *bla* _AIM-1_ *bla* _IMP_ *bla* _IMP-11_ *bla* _IMP-12_	*bla* _NDM_ *bla* _NDM-1_ *bla* _NDM-4_ *bla* _NDM-5_ *bla* _NDM-7_ *bla* _VIM_ *bla* _OXA-48_ *bla* _OXA-181_	PCR, Seq	Cabinda (Angola) Central African RepublicEquatorial Guinea
**Cameroon**	1	*bla*_AIM-1_, *bla*_IMP-11_, *bla*_IMP-12_, *bla*_NDM-1_, *bla*_NDM-4_	PCR, Seq
**Chad**	1	*bla*_NDM-5_, *bla*_OXA-181_	PCR, Seq
**Democratic Republic of Congo**	1	*bla*_KPC_, *bla*_IMP_, *bla*_NDM_, *bla*_VIM_, *bla*_OXA-48_	PCR, Seq
**Gabon**	2	*bla*_NDM-5_, *bla*_NDM-7_, *bla*_OXA-48_	PCR, Seq
**Republic of Congo**	1	*bla* _OXA-181_	PCR, Seq
**São Tomé and Príncipe**	1	*bla* _OXA-181_	PT, PCR, PFGE, Seq
**Western Africa**	**Benin**	2	*bla* _GES_	*bla* _GES_ *bla* _KPC_ *bla* _IMP_ *bla* _NDM_ *bla* _NDM-1_ *bla* _NDM-5_ *bla* _NDM-7_ *bla* _VIM_	*bla* _OXA-48-like_ *bla* _OXA-48_ *bla* _OXA-58-like_ *bla* _OXA-58_ *bla* _OXA-64_ *bla* _OXA-65_ *bla* _OXA-71_ *bla* _OXA-98_ *bla* _OXA-181_ *bla* _OXA-232_ *bla* _OXA-241_ *bla* _OXA-244_	Seq	Cote D’IvoireGuineaGuinea-BissauLiberiaMauritaniaNigerThe GambiaTogo
**Burkina Faso**	2	*bla*_GES_, *bla*_IMP_, *bla*_NDM_, *bla*_VIM_, *bla*_OXA-48-like_, *bla*_OXA-58-like_	Lat. fl., Seq
**Cape Verde**	1	*bla*_OXA-48_, *bla*_OXA-181_, *bla*_OXA-244_	PCR, Seq
**Ghana**	6	*bla*_KPC_, *bla*_NDM_, *bla*_NDM-1_, *bla*_OXA-48-like_, *bla*_OXA-48_, *bla*_OXA-181_	SM, PT, PCR, Seq
**Mali**	2	*bla*_NDM-1_, *bla*_OXA-181_	PT, PCR, Seq
**Nigeria**	11	*bla*_GES_, *bla*_NDM_, *bla*_NDM-1_, *bla*_NDM-5_, *bla*_NDM-7_, *bla*_VIM_, *bla*_OXA-48_, *bla*_OXA-181_, *bla*_OXA-232_	SCM, PT, Lat. fl., PCR, Seq
**Senegal**	1	*bla* _OXA-48_	PT, PCR
**Sierra Leone**	1	*bla*_OXA-58_, *bla*_OXA-64_, *bla*_OXA-65_, *bla*_OXA-71_, *bla*_OXA-98_, *bla*_OXA-241_	PCR, Seq
**Sub-Saharan Africa**	-	2	*bla*_KPC_, *bla*_NDM_, *bla*_VIM_, *bla*_VIM-2_, *bla*_OXA-48_, *bla*_OXA-181_, *bla*_OXA-232_	*bla* _KPC_ *bla* _NDM_ *bla* _VIM_ *bla* _VIM-2_	*bla* _OXA-48_ *bla* _OXA-181_ *bla* _OXA-232_	PT, PCR, Seq

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
