# Peer review of "The Importance of Carbapenemase-Producing Enterobacterales in African Countries: Evolution and Current Burden"

_antibiotics, 2024, doi:10.3390/antibiotics13040295_

Round 1

Reviewer 1 Report

Comments and Suggestions for Authors

Antimicrobial resistance is a public health problem and every day there are bacteria with different resistance mechanisms. These types of reviews are important because we know what is happening in other countries and continents, which allows us to look for strategies to avoid the horizontal transfer of genes that encode for this purpose. The movement of people from one country or continent to another influences the transmission of these resistant bacteria.the English should be revised and the name enterobacterales is in italics.

Author Response

Esteemed Reviewer,

            We, the Authors, would like to thank the Reviewer for their kind words and suggestions. Following the recommendations made, the authors checked the manuscript for typos or grammar mistakes, and sections were rephrased. However, it should be noted that the word Enterobacterales corresponds to the order and, according to taxonomic regulations, would not be written in italics (we learned that some authors use italics for Enterobacterales; please kindly agree to not modify it).

With thanks and best regards,
The Authors

Reviewer 2 Report

Comments and Suggestions for Authors

The authors worked hard to come up with the review article with 21 pages and 195 references. The topic in this manuscript (CPE) is extremely interesting and highly related to this Journal. However, this reviewer has trouble to follow up with what is being described in this article. In such a long review article, there is zero figure and zero table. Many references are cited but no comparisons were made for any prevalence among these countries.

The authors are encouraged to significantly revise this manuscript so that the take-home message is obvious for the readers. One table can be generated to highlight/compare the AMR/CPE among different countries; ideally, statistical analysis can be performed to compare the prevalence of AMR, CPE or in different types of bacterial strains. A review article should have the take-home message and carry its own analysis, rather than the pile-up of the reference.

Also, the authors shall pay more attention to the language. Particularly, the title and abstract needs to be well-written. 1) The title has the key word EVOLUTION, and please be sure the reviewed article has information to do with “evolution”; 2) I had a look at the abstract: Lines 17-18: please revise this sentence. What does “turn pathogenic under certain conditions” mean? Line 24: what does “the past 5-8 years” mean? 5-8 years after 2000s? Or 5-8 years before 2024? Please indicate the years here.

Author Response

Esteemed Reviewer,

Herein we, the Authors, will provide our replies to the commentaries provided in the first round of review for our manuscript. We hope that our response is consistent with the Reviewer’s expectations and that it answers the concerns raised prior.

For ease of reference, we will be including the Reviewer’s notes using quotation marks and italics with our response detailed below the quote. Quotations from the updated manuscript are instead highlighted through the use of bold.

As our manuscript has suffered some changes since its initial submission, the location and positioning of some lines of text have changed; as such, for our response, we will be referencing their position in the updated manuscript .pdf file.

Respectfully,
The Authors

”The authors worked hard to come up with the review article with 21 pages and 195 references. The topic in this manuscript (CPE) is extremely interesting and highly related to this Journal. However, this reviewer has trouble to follow up with what is being described in this article. In such a long review article, there is zero figure and zero table. Many references are cited but no comparisons were made for any prevalence among these countries.”

            We would like to thank the reviewer for their suggestions that are really useful for the betterment of the article. Towards this end, we included a flowchart in the results section of the paper in order to make the manuscript selection process easily readable and provided two tables that offer an overview of the carbapenemase encoding genes associated with Enterobacterales reported in different African regions (Table 1) and an in-depth comparisons between the countries and the results (supplementary Table S_1).

”The authors are encouraged to significantly revise this manuscript so that the take-home message is obvious for the readers. One table can be generated to highlight/compare the AMR/CPE among different countries; ideally, statistical analysis can be performed to compare the prevalence of AMR, CPE or in different types of bacterial strains. A review article should have the take-home message and carry its own analysis, rather than the pile-up of the reference.”

The suggestions made by the reviewer were much appreciated and appropriate changes were made in order to emphasize a clearer take-home message. The supplementary (Table_S1) was created in order to allow the reader to compare all relevant information regarding the topics of high-level AMR, CPE etc. The heterogeneity of data however does not allow proper statistical analysis to be performed, thus we have modified the limitations of the study to further clarify the issue. The text has now been updated to (Lines: 665-670):

”Several other limitations regarding this review were identified:

  • Although significant progress has been made towards implementing efficient measures of surveillance and control, follow-up data is still lacking. When available, the scarcity and heterogeneity of studies hinders the prospects of researchers to properly analyze data. Due to this limit, it was not always possible to specify data important for this review (e.g number of CRE, CPE, genetic profile regarding carbapenemase-encoding genes etc.), as it was uncertain, not available or not applicable (N/A);
  • As English publications were mainly accessed in order to write this review, studies presenting relevant information published in French or other languages might have been overlooked;

”Also, the authors shall pay more attention to the language. Particularly, the title and abstract needs to be well-written. 1) The title has the keyword EVOLUTION, and please be sure the reviewed article has information to do with “evolution”; 2) I had a look at the abstract: Lines 17-18: please revise this sentence. What does “turn pathogenic under certain conditions” mean? Line 24: what does “the past 5-8 years” mean? 5-8 years after 2000s? Or 5-8 years before 2024? Please indicate the years here.”

            We are thankful to the reviewer for pointing out a potential shortcoming of the manuscript. Following the recommendations made, the authors have checked the manuscript for omitted typos or grammar mistakes. The title was rewritten, now reading:  The Importance of Carbapenemase Producing Enterobacterales in African Countries: Evolution and Current Burden.

Regarding the use of the word ”Evolution”, we consider that by presenting the results of our research findings in a chronological manner, the reader can evaluate the improvements of the surveillance programmes as well as steps towards prevention and control. One positive example of evolution is the emergence of prevalence studies for CPE and CRE carriage, while one negative example of evolution is the alarming trends observed by some of the researchers involved in surveilling the AMR phenomenon.

The abstract has been modified in line with the recommendation made.

Lines 17-18 (now 16-17): the sentence was rewritten in order to avoid confusion or ambiguity.;

Line 24 (now 25): the words ”in the past 5-8 years” were replaced with ”after 2015-2018” to better highlight the time period.

Reviewer 3 Report

Comments and Suggestions for Authors

I have read with interest the manuscript submitted by Chelaru et al since AMR represents a global concern.

The manuscript submitted by the authors is very well-structured and wishes to present the full picture of CPE in Africa. The element of novelty, compared to other studies is that it presents the whole picture, for the entire African continent. The reference list is significant and relevant.

My only remark would be the lack of tables/figures, which would significantly improve the interest of readers and would make the article easier to follow/understand.

Best regards,

The reviewer

Author Response

Esteemed Reviewer,

            We, the Authors, would like to thank the Reviewer for the kind words. The suggestion made was much appreciated. Following the recommendations, we included a flowchart in the results section of the paper in order to make the manuscript selection process easily readable and provided two tables: Table 1, that offers an overview of the reported carbapenemase encoding genes associated with Enterobacterales which were reported in different African regions, and a supplementary table (Table_S1), that offers an in-depth comparison between the countries and the studies results pertaining to each of the aforementioned states.

Best regards,
The Authors

Reviewer 4 Report

Comments and Suggestions for Authors

Se enclosed

Author Response

Esteemed Reviewer,

Herein we, the Authors, will provide our replies to the commentaries provided in the first round of review for our manuscript. We hope that our response is consistent with the Reviewer’s expectations and that it answers the concerns raised prior.

For ease of reference, we will be including the Reviewer’s notes using quotation marks and italics with our response detailed below the quote. Quotations from the updated manuscript are instead highlighted through the use of bold.

As our manuscript has suffered some changes since its initial submission, the location and positioning of some lines of text have changed; as such, for our response, we will be referencing their position in the updated manuscript .pdf file.

Respectfully,

The Authors

”Reviewing Report :

The Impact of Carbapenemase Producing Enterobacterales in 2 African Countries : Evolution, Current Burden and Importance of Colonizations

This study, explores a global health problem which is the rise of carbapenemase producing Enterobacterales in African countries. It suggests an investigation into the historical development, present challenges, and the significance of microbial colonizations. However, the study should ensure relevance and focus by avoiding information unrelated to the specified keywords. Also, the study should ensure relevance and focus by avoiding information unrelated to the specified keywords. In general, there are major methodological, and results presentation issues which need to be addressed before the manuscript is re-considered for a next review.”

            We are extremely thankful for the thorough report sent by the reviewer, as we view it as an opportunity to improve the quality of the article. Attentive analysis was performed and information found to be unrelated to the topic was adapted or excluded from the text.  

⮚ Title

The title was clear on the resistance patterns and colonization aspects of Carbapenemase-Producing Enterobacterales in African countries. However, some words in the title, such as "Impact" and "Importance of Colonizations," may lack specificity. While the term "Impact" is used, the study does not describe the sociological, clinical, or economic consequences of carbapenemase resistance. Moreover, the words "Importance of Colonizations" are somewhat ambiguous. We suggest to the authors to rewrite the title.”

We are thankful to the reviewer for pointing out a potential shortcoming of the manuscript. The title was rewritten, now reading: The Importance of Carbapenemase Producing Enterobacterales in African Countries: Evolution and Current Burden.

⮚ Abstract

- Line 15-21 : Introductive sentences is too long and study objective was missing”

We thank the Reviewer for their suggestions. The Abstract introduction was reduced and the objective was added. The revised introduction sentence now reads:

 ”Antimicrobial resistance (AMR) is a worldwide healthcare problem. Multidrug-resistant organisms (MDRO) have the ability to can spread quickly owing to their resistance mechanisms. Although colonized individuals are crucial for MDRO dissemination dispensers, colonizing microbes have the potential to turn pathogenic under certain conditions, leading to symptomatic infections in carriers. Carbapenemase-producing Enterobacterales (CPE) are among the most important MDRO involved in infections and colonizations, associating with multiple resistance mechanisms and virulence factors, causing infections with severe outcomes. All research papers identified in the most comprehensive online databases which contained information related to the topic of this article were analyzed, and relevant data was extracted and included can lead to symptomatic infections in carriers. Carbapenemase producing Enterobacterales (CPE) are among the most important MDRO involved in colonizations and infections with severe outcomes. This review aimed to track down the first CPE reports in Africa, describe their dissemination throughout African countries, and summarize the current status of CRE and CPE data, highlighting current knowledge and limitations of reported data. Two database queries were undertaken using the Medical Subject Headings (MeSH) employing relevant keywords for identifying articles with the topic of beta-lactamases, carbapenemases, carbapenem-resistant, pertaining to Africa or African regions and countries.  The first information on CPE could be traced back to the mid-2000s, but pertinent data for many African countries was established in the past 5-8 years after 2015-2018…”

”- Line 21-22 : Search strategy should be more explained”

            The Reviewer has aptly pointed out that the methodology section of the abstract was lacking. This was updated to reflect actual methodology and some of the search terms. Still, due to constraints regarding the length of the abstract, this was further explained in depth in the Methods section of the article.

”- Line 22-28 : CPE Colonisation rates for countries are missing as well as key data about evolution of the carriage”

Available data is very heterogeneous and it would be rather misleading to discuss in such a limited fashion, as the abstract allows. To this end, the Results, newly-created Table 1, supplementary table (Table S_1) and Discussion were updated to reflect the complexity of the results.

⮚ Introduction

In general the introduction is full of basics and unnecessary informations and the relevance of this study is missing.

- Line 53-58 : Please remove this paragraph (basics informations which are already well known)”

The paragraph was removed, and in its place, a shorter sentence was inserted to highlight only relevant information.

”- Line 74-82 : Please remove this part and replace it by explaining the relevance of this review and the objective.”

            The text was removed and an explanation regarding the relevance of this review and the objective were added.

⮚ Method

- Please precise exact period of litterature research”

            Considering the availability of data in certain regions and scarcity in other, an exact period was determined to be detrimental to the study as a whole. Thus, the authors concluded that the period of literature research should be referred to as ”until October 2023”.

”- The key words used for this research is insufficient, please add some keywords and put them in a table which each database in a column”

            We want to thank the reviewer for pointing out a potential gap within our methodology. We consider the chosen keywords as being inclusive for all relevant research within the field. We would kindly ask the reviewer to point out which keywords would bring further value to our work.

”- Please make a flow diagram of the selection of the studies and explain all steps for selection”

            We would like to thank the reviewer for their suggestions. Towards this end, we updated the methodology and included a flowchart in the results section of the paper in order to make the manuscript selection process easily readable.

⮚ Results

In general, the author should specify the origins of the isolates (samples) to distinguish between colonization and infection. Throughout the entire document, the author discusses carriers without explicitly specifying whether it pertains to digestive carriage, nasal carriage, or another type. Additionally, I suggest for the authors to include a table providing detailed information on bacterial species, sample types, detection methods, collection dates, countries, and references for each study. This would contribute to a clearer presentation of the research data.

            We thank the reviewer for pointing out the vagueness of certain information and for the offered suggestions. In order to better describe the studies we provided a supplementary table (Table_S1), that offers an in-depth comparison between the countries and on data reported in the studies. Also, explanations were included throughout the text.

”- Line 109 : Which traits explain the similarity between the isolates ?”

  • Nordmann et al. analyzed the pulse-field gel electrophoresis patterns of the isolates. The supplementary information was included in the text. (now Line 144)

”- Line 115 : Bla GES gene is a known beta-lactamase gene that confers resistance to beta-lactam antibiotics, including penicillins and cephalosporins. It is not a carbapenemase producing gene.”

  • The phrase was reformulated and an explanation regarding the blaGES genes was added in the Discussion. Regarding GES-2, we would like to respectfully point out that it is a carbapenemase (http://www.bldb.eu/BLDB.php?prot=A#GES). (now Line 150)

”- Line 146 : The study describe results about colonizations, please explain why environmental samples such as wastewater are described”

  • As data regarding CR or CP Enterobacterales was very scarce for some countries, environmental data was included in order to emphasize the fact that such strains or CP genes are present in those countries. Environmental contamination can be a source of colonization and infection. (now Line 184)

”- Line 147 : Colonization refers to the presence of bacteria without causing harm or symptoms. I suggest to clarify this point since in this study urinary tract infection was documented.”

  • One of the scopes of our paper refers to the current burden of CPE and CRE. Thus, we deemed it relevant to include the data indicated. (now Line 185)

”- Line 150 : Authors should remove this information if no data on CP is available for the study ”

  • Considering the scarcity of available data, we deemed it relevant to note that no further data was available, thus enforcing the quality of the available information in some countries. (now Line 188)

”- Line 298 : Positive blood culture refers to septicemia (infection) which is different from colonization”

  • One of the scopes of our paper refers to the current burden of CPE and CRE. Thus, we deemed it relevant to include the data indicated. (now Line 348)

⮚ Discussion

The discussion mentions reports of highly resistant CPE in the past 5-6 years, but it lacks a deeper exploration of the factors contributing to the rise in resistance, including the role of antibiotic prescriptions and accessibility of testing techniques.”

The situation causing CPE spread in Africa may be due to a series of factors that cannot be reliably explored in the current review. Challenges related to the social and political instability in certain African regions, are beyond the scope of this study.

”- Please remove the first paragraph and write general sentence about carbapenem resistance and its carriage”

An introductory, general part was written.

”- Line 485 : How can the authors be sure that the first strains emerged in the mid or early 2000s in Africa since emergence of carbapenem resistance was reported before accross the world. I suggest to add « the first first strains emerged was reported in the mid or early 2000s »”

We would like to thank the reviewer for identifying a misuse of terms. We adapted the manuscript accordingly and it now reads:

”It is hard to say with certainty when, where or how the CPE began to spread in Africa, as there are many factors involved, but the first studies describing the emergence of CPE evaluated strains isolated in the mid or early 2000s. These strains were disseminated through various ways, including asymptomatic carriers. It should be noted that some studies reported data from the same year the study was published, while others from previous years.”

⮚ Conclusion

The conclusion should be rewritten. The conclusion should succinctly recap the main findings of the review, emphasizing the critical points related to the emergence, spread, and impact of Carbapenemase Producing Enterobacterales (CPE) in African countries. This provides readers with a quick overview of the essential insights.”

            We thank the reviewer for pointing out the inconsistencies and for offering suggestions regarding this part of the article. The conclusions have been rewritten accordingly.

Round 2

Reviewer 2 Report

Comments and Suggestions for Authors

This reviewer appreciated the hard work from the authors in revising this manuscript. The newly added table makes this review article much easier to follow.

I would suggest to further modify/simplify the title of this article. How about this one: “Evolution and Prevalence of Carbapenemase-Producing Enterobacterales in Africa”.

Also, the authors are encouraged to carefully go though the entire manuscript and make sure the format of writing the names of bacteria and AMR gene is correct.sssThis reviewer appreciated the hard work from the authors in revising this manuscript. The newly added table really makes this review article much either to follow. I would suggest to further modify/simplify the title of this article. How about this one: “Evolution and Prevalence of Carbapenemase-Producing Enterobacterales in Africa”. Also, the authors are encouraged to carefully go though the entire manuscript and make sure the format of writing the names of bacteria and AMR gene is correctThis reviewer appreciated the hard work from the authors in revising this manuscript. The newly added table really makes this review article much either to follow. I would suggest to further modify/simplify the title of this article. How about this one: “Evolution and Prevalence of Carbapenemase-Producing Enterobacterales in Africa”. Also, the authors are encouraged to carefully go though the entire manuscript and make sure the format of writing the names of bacteria and AMR gene is correctThis reviewer appreciated the hard work from the authors in revising this manuscript. The newly added table really makes this review article much either to follow. I would suggest to further modify/simplify the title of this article. How about this one: “Evolution and Prevalence of Carbapenemase-Producing Enterobacterales in Africa”. Also, the authors are encouraged to carefully go though the entire manuscript and make sure the format of writing the names of bacteria and AMR gene is correctThis reviewer appreciated the hard work from the authors in revising this manuscript. The newly added table really makes this review article much either to follow. I would suggest to further modify/simplify the title of this article. How about this one: “Evolution and Prevalence of Carbapenemase-Producing Enterobacterales in Africa”. Also, the authors are encouraged to carefully go though the entire manuscript and make sure the format of writing the names of bacteria and AMR gene is correct

Author Response

Esteemed Reviewer,

We, the Authors, would like to thank the Reviewer for the suggestions provided in the second round of review for our manuscript.

Following the recommendations made, the manuscript was carefully checked and the format / names of bacteria and AMR genes were corrected / adapted accordingly.

Regarding the suggestion towards modifying the title, the authors would appreciate if you would kindly accept that the manuscript be accepted for publication under the current title.

With thanks and best regards,
The Authors

Reviewer 4 Report

Comments and Suggestions for Authors

All my concerns have been well adressed. The paper could be accepted in present form.

Author Response

Esteemed Reviewer,

We, the Authors, would like to thank the Reviewer for the provided feedback.

With best regards,
The Authors